# A Comprehensive Study of the Wave Impact Loads on an Inclined Plate

**Zhe Ma [1,2], Ting Zhou [2], Nianxin Ren [1,2,*] and Gangjun Zhai [1,2]**

[1]  State Key Laboratory of Coastal and Offshore Engineering, Dalian University of Technology, Dalian 116024, China; deep_mzh@dlut.edu.cn (Z.M.); zhai@dlut.edu.cn (G.Z.)

[2]  Deepwater Engineering Research Centre, Dalian University of Technology, Dalian 116024, China; deep_zhouting@mail.dlut.edu.cn

*  Correspondence: rennianxin@dlut.edu.cn

**Abstract:** Water wave impact on a wet deck is an important issue in ocean engineering, and the plate-shaped structure in the splash zone tends to suffer tremendous impact loads. This work presents a method for predicting the wave slamming uplift force on a fixed plate with different inclined angles. Both numerical simulation and the scale model test of the wave impact loads on an inclined plate were performed, and a good agreement was obtained. In addition, the influence of three important wave parameters on the slamming uplift force was systematically investigated: relative deck width $B/L_S$, relative wave height $\Delta h/H_{1/3}$, and the plate's inclined angle $\alpha$. The results indicate that the three parameters can significantly influence the wave slamming uplift force. Finally, a developed empirical equation is proposed for estimating the wave slamming uplift force on the inclined plate.

**Keywords:** wave impact; inclined plate; model tests; empirical equations

## 1. Introduction

As an important renewable energy source, marine energy is playing a vital role in solving the growing shortage of land energy and facilitating the necessary transition to clean and low-carbon energy. Over the past 40 years, various kinds of marine structures have been established as effective platforms for renewable energy extraction. Meanwhile, wave slamming is a phenomenon that occurs when marine structures encounter hostile ocean environments. The pressure loads induced by slamming are characterized by a large magnitude and short duration [1], so it affects the structure both locally and globally, causing considerable structural damage. Therefore, the precise prediction of impact loads acting on a structure has become a crucial problem for researchers in the pre-design of marine structures.

Studies on wave slamming were originally conducted by researchers in the fields of ships and aeronautics, and the wave entry problem has become a hot topic throughout the years. A pioneer study on this subject resulted in the development of a theoretical solution of the sea-plane landing problem, and a wedge-shaped model was utilized on the basis of momentum theorem and water added mass assumptions [2]. Problems related to water entry of a solid body with a relatively low deadrise angle [3], as well as water wave impact pressures on the wetted deck, green water, tank sloshing, etc. [4,5], have been previously investigated. An experimental investigation of the pressure distribution on wedge sections with different deadrises during water entry was conducted [6]. Analytical and volume-of-fluid (VOF) methods were utilized to study the water entry problem with a circular cylinder [7]. Numerical models were established to study the water entry problem for different velocities and directions by using the smoothed particle hydrodynamics (SPH) method [8], and the model successfully yielded the optimal prediction of object movement. The time-domain higher-order boundary element method (HOBEM) was applied [9,10] to investigate the hydrodynamic

performance of wave slamming when an oscillating wave surge converter (OWSC) interacts with the waves accompanied by the effects of current. Studies have proved that the compressibility of the water, air cushions, air bubbles, and even hydroelasticity can have an influence on the issue, making the process of wave slamming very complex [11].

Various methods have also been developed to investigate the interaction between fixed structures and waves both theoretically and experimentally. Although it is different from the wave entry problem to some extent, the methods are universal and widely adopted. Laboratory experiments were carried out to investigate the issue. Global impact loads on a fixed and rigid horizontal wet deck were studied for regular incident waves and 2D-flow conditions [12]. Local hydroelastic wet deck slamming effects were studied on the Ulstein Test Catamaran [13]. A foundational study on this subject used physical model tests to investigate wave loads on horizontal decks that were subject to breaking and non-breaking wave attacks [14], and results revealed a substantial similarity between the mechanisms of wave impact on horizontal platforms and those on vertical barriers. Experiments on the impact loads on a flat plate were conducted and generated key findings on trapped air and its effect on impact pressure [15]. In recent years, another significant study investigating water wave impact was done by taking the elasticity of the plate into consideration [16]. Further developments on wave impact have been discussed by others [17] who have investigated the transient properties of the flow field of a wave slamming a superstructure of an open structure, and the instantaneous velocity field of the impacting wave was obtained using a particle image velocimetry (PIV) system.

Most of the investigations above have illustrated the basic theory underlying the slamming of the water–plate interaction. However, few systematic studies have examined the effect of inclined angles on the distribution of the wave uplift force. For floating marine structures, inclined angles actually exist in response to waves, and harsh wave conditions and large inclined angles always coincide. However, wave slamming itself is a complex process accompanied by bubbles and compressibility problems. In addition, the dynamic response of floating platforms makes it a challenge to study this problem. Therefore, the main aim of this article is to study the wave uplift force on a fixed plate with inclined angles attributed to the impact of a massive wave. Considering the fact that wave slamming occurs instantaneously, applying a fixed plate with different angles can also contribute to the investigation of the slamming phenomenon by focusing on the significant effect of inclined angles on the evolution of bubbles and the slamming force, which happens within a very short time.

The paper is structured as follows. First, in Section 2, the experimental set-up and wave generation for irregular waves is introduced. In addition, the governing equations for the numerical model are presented, followed by a description of the computation domain and boundary conditions. In Section 3, both regular waves and irregular waves are applied to systematically study the influence of three interdependent parameters involved in the slamming problem: the width of the plate compared with the wave length, the air gap compared with the wave height of the incident waves, and lastly, the inclined angle of the plate. Based on the physical results, an empirical equation evaluating the wave uplift force is presented, and its precision is examined. Conclusions are summed up in Section 4 to enclose the paper.

## 2. Methods

### 2.1. Experimental Set-Up

Experiments were performed in a narrow wave tank in the Research Institute of Ocean Engineering, Dalian University of Technology. The wave tank was 50.0 m long, 1.0 m wide, and 1.5 m high and was equipped with a plunge-type wave generator at one end and a wave absorbing device (a sloped beach covered with rocks and absorbent matting) at the other end. The schematic of the experimental set-up is shown in Figure 1a. A simple box-shaped geometry was used. The deck model, made of organic glass, was 0.5 m long, 0.5 m wide, and 0.01 m thick; thus, elastic deformations could be neglected in subsequent experiments. The model was installed in the center of the tank, 0.25 m away from the edge

of the plate to the side walls of the tank. The clearance Δh (the distance from the bottom deck to the still wave surface) was constant at 0.03 m. The model was connected to a supporting structure, with one end fixed to the model deck and the other to a bracket placed on the top of the tank. With the device, the plate-shaped model could be adjusted to the proper angle α and then fixed (see Figure 1b). Twenty-five pressure transducers were fixed on the underside of the deck model and marked from 1 to 25, as shown in Figure 2. PT1 to PT5 in Figure 1b are 5 pressure transducers which correspond to No. 11–15 pressure sensors, respectively. In the preparation stage in the experiment, to obtain the wave surface elevation and then check whether the measured wave height meets the desired one in the wave tank, a wave gauge was arranged, which was 24.15 m away from the wave maker. The wave impact pressure on the underside of the deck was measured using a DS-30 multi-point pressure measuring system, which is made by the Institute of Water Transportation of Tianjin, China. The frequency of the pressure transducers has a maximum of 3000 Hz. For regular waves, the frequency was 1000 Hz, with 0.001 s as the data-sampling interval; for irregular waves, the frequency was 500 Hz, with 0.002 s as the data-sampling interval.

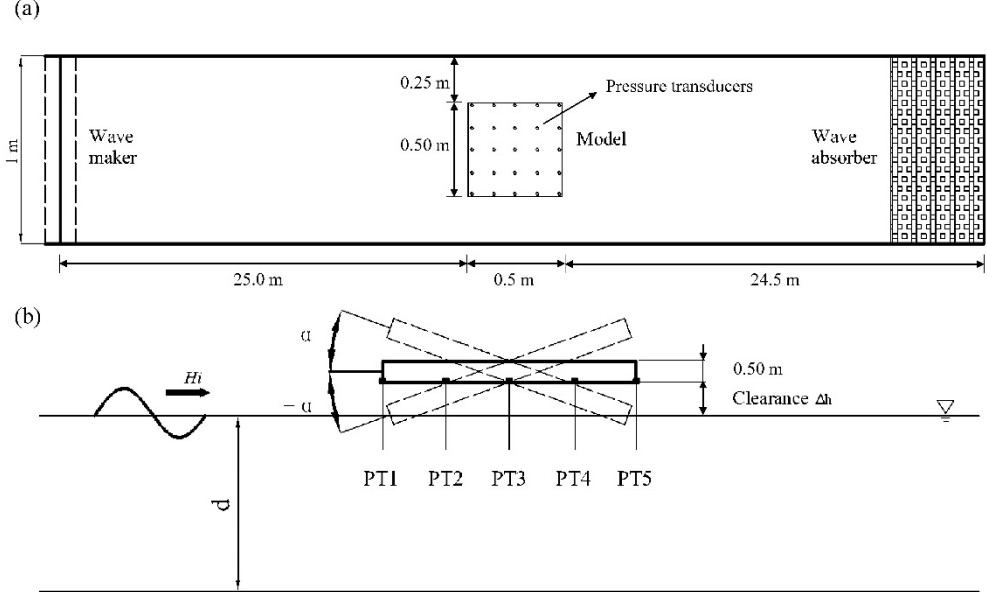

**Figure 1.** Sketch of small-scale experimental set-up: (**a**) the top view; (**b**) the front view.

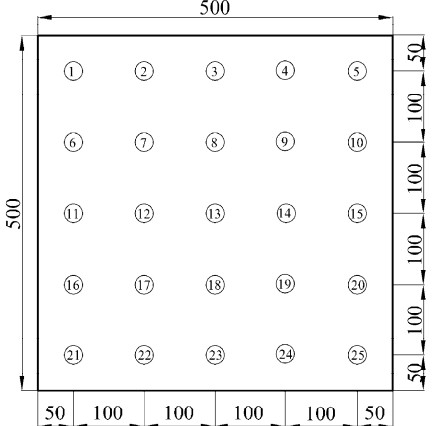
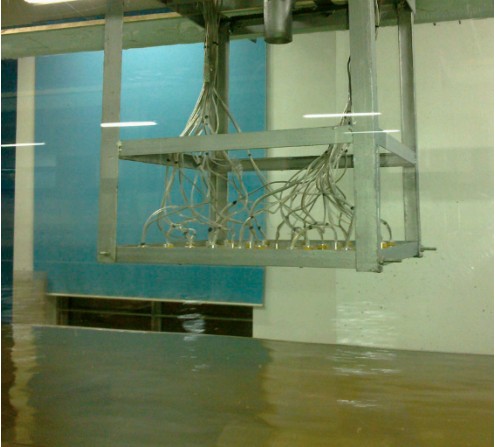

**Figure 2.** Sketch of the distribution of pressure transducers on the surface of the deck model (unit: mm).

### 2.2. Irregular Wave Generation

A modified Jonswap spectrum was chosen as the target spectrum. In linear wave theory, the initial free surface elevation can be represented by a linear combination of sinusoidal waves.

$$\eta(t) = \sum_{i=1}^{n} \sqrt{2S(f_i) \cdot df} \cos(2\pi \cdot f_i \cdot t + \tau_i) \tag{1}$$

where $\eta(t)$ is the free surface elevation of the initial waves, and $\tau_i$ is the phase constant, which is given by uniformly distributed random numbers. The modified Jonswap spectrum can be expressed as follows (Goda, 1999 [17]):

$$S(f) = \beta_j \cdot H_{1/3}^2 \cdot T_p^{-4} \cdot f^{-5} \cdot \exp\left[-\frac{5}{4} \cdot (T_p \cdot f)^{-4}\right] \cdot \gamma^{\exp\left[-\left(\frac{f}{f_p}-1\right)^2/2\sigma^2\right]} \tag{2}$$

$$\beta_j = \frac{0.06238(1.094 - 0.01915 \ln \gamma)}{0.23 + 0.0336\gamma - 0.185(1.9 + \gamma)^{-1}} \tag{3}$$

$$T_p = \frac{T_{1/3}}{1.0 - 0.132(\gamma + 0.2)^{-0.559}} \tag{4}$$

$$\sigma = \begin{cases} 0.07 & f \le f_p \\ 0.09 & f > f_p \end{cases} \tag{5}$$

where $H_{1/3}$ and $T_{1/3}$ are the significant wave height and period, respectively; $f$ is the frequency; $\gamma$ is the peak enhancement factor, $\gamma = 3.3$; $T_p$ and $f_p$ denote the wave period and the frequency at the spectral peak, respectively; and $\sigma$ represents the shape parameter.

The water depth $d$ was constant at 0.6 m. The significant wave height $H_{1/3}$ was set to 0.06 m, 0.08 m, 0.10 m, and 0.12 m, and the peak period of the wave spectrum $T_p$ was set to 1.0 s, 1.2 s, 1.4 s, 1.6 s, 1.8 s, and 2.0 s. The characteristic wave length $L_s$ was 1.538 m, 2.122 m, 2.705 m, 3.271 m, 3.822 m, and 4.361 m, respectively, and it was calculated on the basis of linear wave theory. The relative deck width $B/L_s$ ranged from 0.115 to 0.325. The relative clearance, defined as the ratio of the clearance of the underside of the structure above the still water level to the significant wave height, $\Delta h/H_{1/3}$, ranged from 0.25 to 0.5, and the plate's inclined angel $\alpha$ ranged from $-10°$ to $10°$. For each case, the time series of wave impact pressures were recorded at least three times by all 25 pressure transducers on the surface of the plate model to mitigate accidental errors that occur during the experiments.

### 2.3. Numerical Model

The interaction between a regular wave and a horizontal plate ($\alpha = 0°$) was simulated using our in-house solver: DUT-FOAM [18]. As a two-phase incompressible Reynolds-Average Navier–Stokes (RANS) solver, DUT-FOAM was developed on the basis of OpenFOAM-4.1.

For the fluid domain, the finite volume method [19] was used to discretize Equations (6) and (7). The volume-of-fluid (VOF) method [20] was utilized for the free surface capture. A complete system, including a wave generation and absorption model, was applied: the mass source wave generation method and linear damping scheme [21] were adopted for the numerical calculations reported in this part.

The continuity, RANS, and VOF equations are as follows:

$$\nabla \cdot \boldsymbol{u} = 0 \tag{6}$$

$$\frac{\partial \rho \boldsymbol{u}}{\partial t} + \nabla(\rho \boldsymbol{uu}) - \nabla \cdot (\mu \nabla \boldsymbol{u} + \rho \tau) = f_\sigma - gX\nabla\rho - \nabla P_{\rho gX} \tag{7}$$

$$\frac{\partial \alpha}{\partial t} + \nabla \cdot \alpha \boldsymbol{u} + \nabla \cdot [\alpha(1-\alpha)\boldsymbol{u_r}] = 0 \tag{8}$$

where $\boldsymbol{u}$ represents the velocity vector; $\alpha$ is the water volume fraction; $\rho\tau$ is the Reynolds stress term, in which $\tau$ represents the Reynolds stress tensor caused by pulsation and is a second-order symmetric tensor; $\rho$ and $\mu$ are the average density and dynamic viscosity of the two-phase flow, respectively, and are calculated by:

$$\rho = \alpha\rho_1 + (1-\alpha)\rho_2 \tag{9}$$

$$\mu = \alpha\mu_1 + (1-\alpha)\mu_2 \tag{10}$$

where $f_\sigma$ in Equation (7) is the surface tension term, in which $C$ is the surface tension coefficient, and $\kappa$ is the curvature of the interface; $g$ is the acceleration of gravity; $X$ is the position vector in Equation (8); $\boldsymbol{u_r}$ represents the relative velocity; and the third term in Equation (8) is the artificial compression term [22], which is introduced to compensate for the numerical smearing of the fluid interface.

The mass source method was used to generate the target wave; thus, a source term is introduced to the continuity Equation (1) in the source region, as shown in Equation (11):

$$\frac{\partial u_i}{\partial x_i} = s(x_i, t) \tag{11}$$

where $s(x_i, t)$ equals a nonzero mass source function within the source region $\Omega$. By assuming that all of the mass increases or decreases introduced by the mass source function contribute to the generation of the target wave, Lin et al. [23] derived a relationship between the source function $s(x_i, t)$ and the expected time history of free surface displacement $\eta(t)$:

$$\int_0^t \int_\Omega s(x, y, t)d\Omega dt = 2\int_0^t C\eta(t)dt \tag{12}$$

where $C$ is the phase velocity of the target wave. Further simplifying Equation (12) results in the following equation:

$$s(x, y, t) = \frac{2C\eta}{A} \tag{13}$$

where A represents the area of the mass source region. Here, $A = Ls \times Hs$, where $Ls$ and $Hs$ are the length and height of the mass source region, respectively. For a linear monochromatic wave, $\eta(t) = H \sin(\omega t)/2$, where $\omega$ represents the wave frequency. By substituting $\eta(t)$ into Equation (13), the final source function is as follows:

$$s(x, y, t) = \frac{CH}{A}\sin(\omega t) \tag{14}$$

Numerical simulations were carried out by using a numerical wave tank (NWT) with the center of the plate placed at $x = 6.2$ m, as depicted in Figure 3. However, several simplifications were made to eliminate the computational cost. First, both ends of the NWT were set as damping zones, ensuring little negative effect on the domain for the wave–plate interaction. Second, a shorter wave tank (12.0 m) than the physical wave tank (50.0 m) was used. Sensors for measuring pressure were installed under the plate, and the position of each measuring pressure relative to the plate were consistent with those in experiment as Figure 2 shown.

The computational domain was discretized by the snappyHexMesh pre-processor. A stationary Cartesian grid was employed, and the grids near the free surface were denser. The maximum cell dimension was 0.02 m in the x- and y-directions and 0.01 m in the z-direction. The minimum cell dimension in the z-direction was 0.005 m, resulting in approximately $3.66 \times 10^6$ cells in total. The time step was auto-adjusted to satisfy the criterion of a maximum Courant number of 0.5, and the whole simulation time was up to 20.0 s. The initial time step was set as 0.001 s. During the whole calculation process, the maximum time step was 0.004 s and 0.0003 s minimum time step.

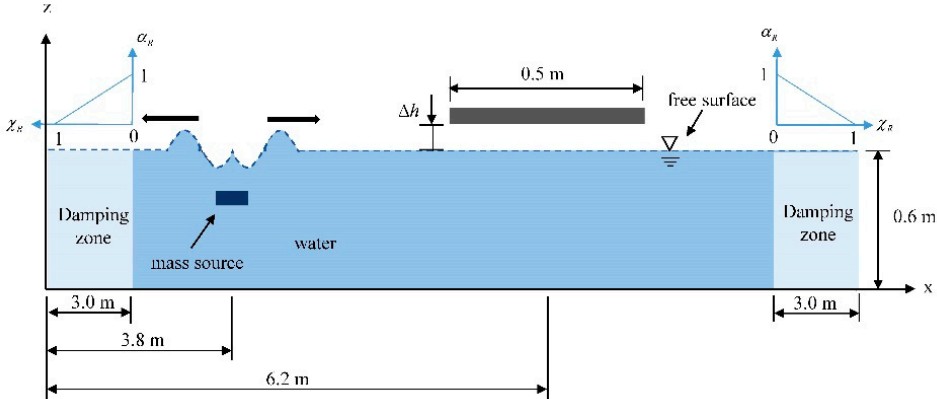

**Figure 3.** Geometric model of the numerical wave tank (NWT).

The boundary conditions applied at the domain were set as follows: (i) $p = 0$ and $(\nabla u)\,\boldsymbol{n} = 0$ on the atmosphere (top) boundary; (ii) free-slip and $\boldsymbol{n} \cdot \nabla p = 0$ on the bottom, left, and right boundaries; (iii) no-slip and $\boldsymbol{n} \cdot \nabla p = 0$ on the plate surface; (iv) symmetry plane boundary conditions on the front and back surface.

## 3. Results and Discussion

In this section, predictions of interactions between the target waves and a fixed plate with inclined angles are presented for both regular waves and irregular waves. Our concerns in this paper are three: target wave generation, distribution of impact load under the deck model, and reliable prediction of the wave uplift force. Experimental measurements are compared with numerical results for regular waves. The effect of wave-in-plate phenomena is studied systematically.

### 3.1. Parametric Studies of the Wave Uplift Force with Regular Waves

In this part, the wave-in-plate interactions for regular waves are reported according to both numerical and experimental results. Wave conditions are as follows: for regular waves, wave height $H$ was from 0.08 m to 0.14 m, wave period $T$ from 1.0 s to 1.8 s, and water depth $d = 0.6$ m. The whole computational time was 20.0 s.

Before the simulation of waves interacting with the plate, tests were performed to verify the accuracy of wave generation by the established NWT in each wave condition. Figure 4 reports a comparison of the computational and analytical time histories of the wave elevations at the 12th point under wave condition of $H = 0.08$ m, $T = 1.0$ s. The relative error of wave height was 5.5%, which illustrates a fairly good agreement between the computation and the analytical results.

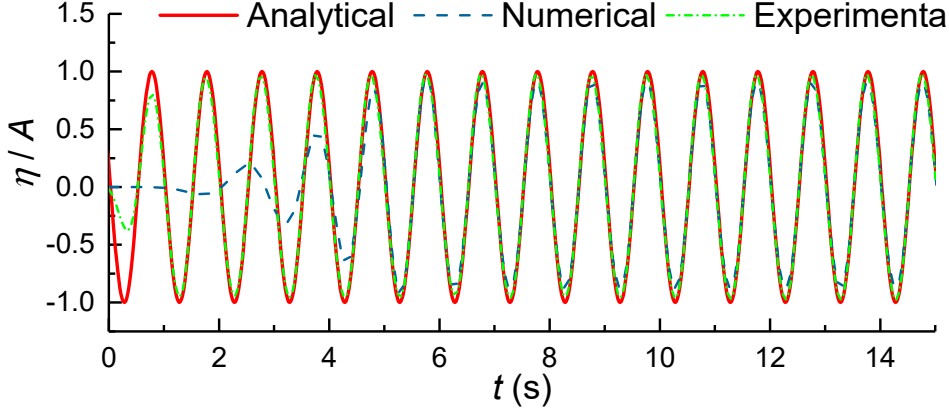

**Figure 4.** A comparison of three different solutions for regular waves with wave height $H = 0.08$ m.

Subsequently, the interaction between regular waves and the plate was calculated numerically, and the time history of the slamming pressure was captured. Taking the 12th pressure measurement as an example (see Figure 5), an initial peak pressure of considerable magnitude but of short duration and a slowly varying uplift pressure of less magnitude but of considerable duration are both revealed in the experimental and numerical results. The magnitudes in positive pressure of numerical results are smaller than experimental ones. It is easy to expect because the generated wave height in NWT is less than the desired one. There is another difference in the negative magnitude: numerical results represent a longer duration than experimental ones. A possible reason for this discrepancy is that for real conditions, such as those in the experiments, as a wave separates from the bottom surface of the plate, the air bubble expands gradually because the pressure in the bubble decreases. However, the numerical model does not take air compressibility into account, causing the negative pressure to last longer in the numerical simulations relative to that observed in the experimental conditions. In general, the results shown in Figure 5 are acceptable and reasonable.

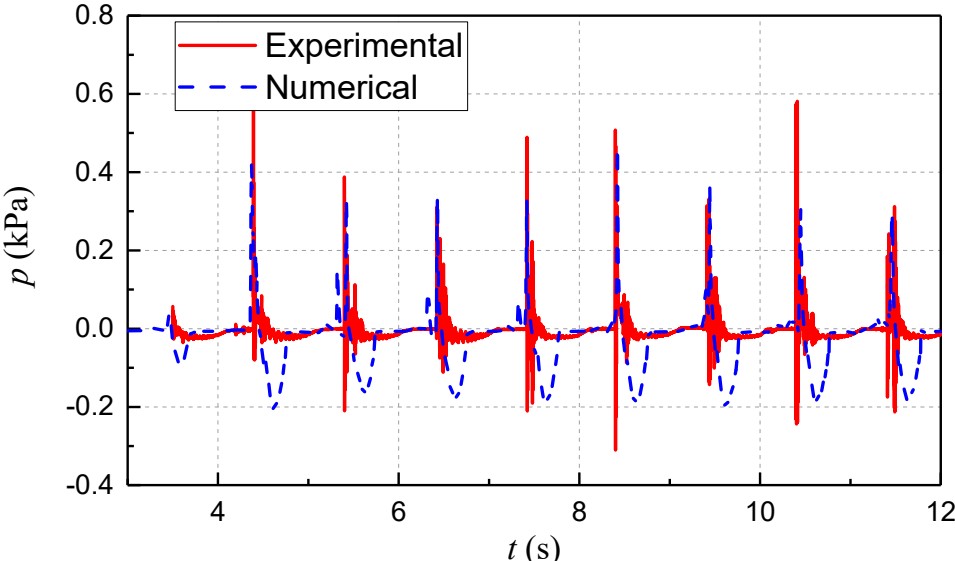

**Figure 5.** A Comparison of slamming pressure on a horizontal plate in the 12th pressure measurement ($H = 0.08$ m, $T = 1.0$ s).

Further analysis was carried out to study the scatter of wave uplift force at different combinations of wave period $T$ and inclined angel $\alpha$. The numerical results are summed up in Table 1. When the inclined angles are negative, the maximum impact peak value reached 8 kPa at PT2, which is a dangerous area requiring extra attention. On the contrary, when the inclined angles are positive, the total pressure level is higher, and we can find the impact peak value emerges in turn at PT1 to PT5. When comparing the experimental and numerical results at each inclined angel, there is a common tendency for the values to change along the five points generally speaking. Therefore, the numerical results are in good agreement with experimental results at most measure points. For further comparison, some interesting phenomena are captured. Take $\alpha = 0°$ as an example, the deviation between the experimental values and the numerical ones is smallest at PT3 for all periods. This can be explained as follows: the closer the pressure measuring point is to the edge of the plate, the more easily the wave breaks up, and complex conditions occur. In this circumstance, the compressibility of air bubbles, and even hydro-elasticity will influence the fields [11]. Therefore, for points near the edge, there is a challenge to obtain the impact pressure accurately by a numerical model, and minor errors between the experimental values and the numerical ones appear. In addition, in other inclined angle conditions, there is a similar phenomenon to discover.

**Table 1.** The maximum impact pressure with respect to the inclined angel $\alpha$ and wave period T for different periods at 5 positions ($H = 0.14$ m).

| Maximum Impact Pressure (kPa) | | T = 1.0 s | | T = 1.2 s | | T = 1.4 s | | T = 1.6 s | | T = 1.8 s | |
|---|---|---|---|---|---|---|---|---|---|---|---|
| Angle | n | N | E | N | E | N | E | N | E | N | E |
| −10° | 1 | 0.72 | 0.71 | 2.56 | 1.59 | 2.47 | 0.49 | 2.32 | 0.24 | 0.85 | 0.49 |
| | 2 | 6.99 | 8.55 | 6.59 | 8.66 | 2.84 | 1.60 | 1.22 | 2.44 | 1.22 | 1.59 |
| | 3 | 1.08 | 2.17 | 0.98 | 1.59 | 0.62 | 1.23 | 1.10 | 0.98 | 0.73 | 0.73 |
| | 4 | 0.36 | 0.72 | 0.73 | 1.59 | 0.49 | 0.49 | 0.85 | 0.24 | 0.49 | 0.37 |
| | 5 | 0.36 | 0.72 | 0.24 | 0.61 | 2.59 | 4.32 | 0.61 | 0.00 | 0.49 | 0.37 |
| −5° | 1 | 4.46 | 2.29 | 2.20 | 1.22 | 2.10 | 1.23 | 1.71 | 0.73 | 0.73 | 0.73 |
| | 2 | 1.81 | 0.84 | 6.22 | 3.41 | 3.95 | 5.80 | 3.54 | 4.88 | 5.00 | 7.07 |
| | 3 | 2.53 | 3.01 | 1.59 | 2.20 | 0.74 | 1.48 | 1.10 | 1.22 | 0.37 | 2.20 |
| | 4 | 0.36 | 2.29 | 1.46 | 1.22 | 0.74 | 1.36 | 1.22 | 0.61 | 0.24 | 0.61 |
| | 5 | 0.00 | 0.72 | 0.12 | 1.10 | 0.12 | 0.86 | 0.49 | 0.37 | 0.24 | 1.10 |
| 0° | 1 | 3.49 | 5.18 | 0.85 | 1.83 | 1.98 | 4.44 | 5.00 | 3.41 | 1.46 | 1.34 |
| | 2 | 0.48 | 0.96 | 1.34 | 1.95 | 4.94 | 4.32 | 5.61 | 3.90 | 1.95 | 1.95 |
| | 3 | 0.72 | 1.33 | 2.68 | 2.20 | 2.10 | 1.98 | 3.05 | 1.34 | 5.24 | 5.85 |
| | 4 | 2.53 | 4.10 | 1.59 | 2.93 | 1.11 | 1.36 | 2.56 | 4.02 | 2.20 | 2.68 |
| | 5 | 0.60 | 0.96 | 0.61 | 1.95 | 3.95 | 4.32 | 1.59 | 3.90 | 1.34 | 2.80 |
| 5° | 1 | 3.49 | 4.58 | 0.73 | 2.32 | 2.59 | 6.54 | 5.98 | 5.12 | 4.27 | 2.93 |
| | 2 | 1.20 | 1.69 | 1.34 | 2.32 | 5.06 | 5.80 | 6.95 | 5.00 | 5.12 | 4.27 |
| | 3 | 1.20 | 1.20 | 2.56 | 2.44 | 5.06 | 3.95 | 4.51 | 1.59 | 4.88 | 4.27 |
| | 4 | 1.81 | 1.45 | 2.56 | 3.29 | 2.47 | 3.46 | 2.93 | 0.61 | 5.00 | 6.34 |
| | 5 | 1.08 | 1.69 | 0.73 | 2.32 | 4.57 | 5.80 | 1.22 | 4.63 | 3.66 | 4.63 |
| 10° | 1 | 4.70 | 3.98 | 3.66 | 2.56 | 3.33 | 2.84 | 3.54 | 1.95 | 1.71 | 2.56 |
| | 2 | 5.42 | 5.06 | 7.07 | 6.22 | 6.30 | 6.42 | 4.39 | 3.29 | 4.15 | 4.88 |
| | 3 | 5.78 | 7.35 | 5.49 | 5.24 | 6.05 | 5.43 | 5.61 | 3.41 | 3.05 | 2.68 |
| | 4 | 5.78 | 6.14 | 7.93 | 8.66 | 4.20 | 3.46 | 3.54 | 1.71 | 5.12 | 3.54 |
| | 5 | 4.82 | 5.06 | 4.39 | 6.22 | 7.16 | 6.30 | 2.44 | 3.17 | 3.78 | 4.88 |

Remarks: $\alpha$ for inclined angel of the plate; n for the 5 positions of pressure measure points (PT1 to PT5); N for numerical results; E for experimental results.

### 3.2. Parametric Studies of the Wave Uplift Force with Irregular Waves

Previous studies have pointed out that if the relative angle between the body and fluid is smaller than 5°, the oscillating air enclosure should have an effect [24]. Therefore, for better research, the experimental wave tank is appropriate for studying water wave impact problem. In this part, the experimental results of wave–plate interactions for irregular waves are presented. The wave conditions are listed in the previous section, so the information is not repeated here. The whole duration is 200.0 s, ensuring at least 100 waves are generated in each wave spectrum. Settings of duration in this way is sufficient to study wave impact and ensure the reliability of results.

First, comparisons of the measured spectrum with the target spectrum for four typical cases are shown in Figure 6. It can be observed that the measured wave spectra are in good agreement with the target spectra for each case. On the basis of impact pressure sampling records of the 25 pressure transducers on the underside of the deck, the wave uplift force of the deck was calculated by a simplified mean integration method. The plate is evenly divided into 25 parts, with each part is a small square (0.1 m × 0.1 m), and the uplift force is the sum of the 25 components. As the pressure sensors are all scattered in the center of the small part and 25 measuring points are totally dense enough for a plate of such size, and the total force $F(t)$ is calculated as follows:

$$F(t) = \sum_{i=1}^{25} f_i(t) \cdot A \tag{15}$$

where, $f_i(t)$ is the measured pressure force over time of each sensor, $i = 1\sim25$, $A$ is the area of each smaller square. The time series of the wave uplift forces for different inclined angles are shown in Figure 7.

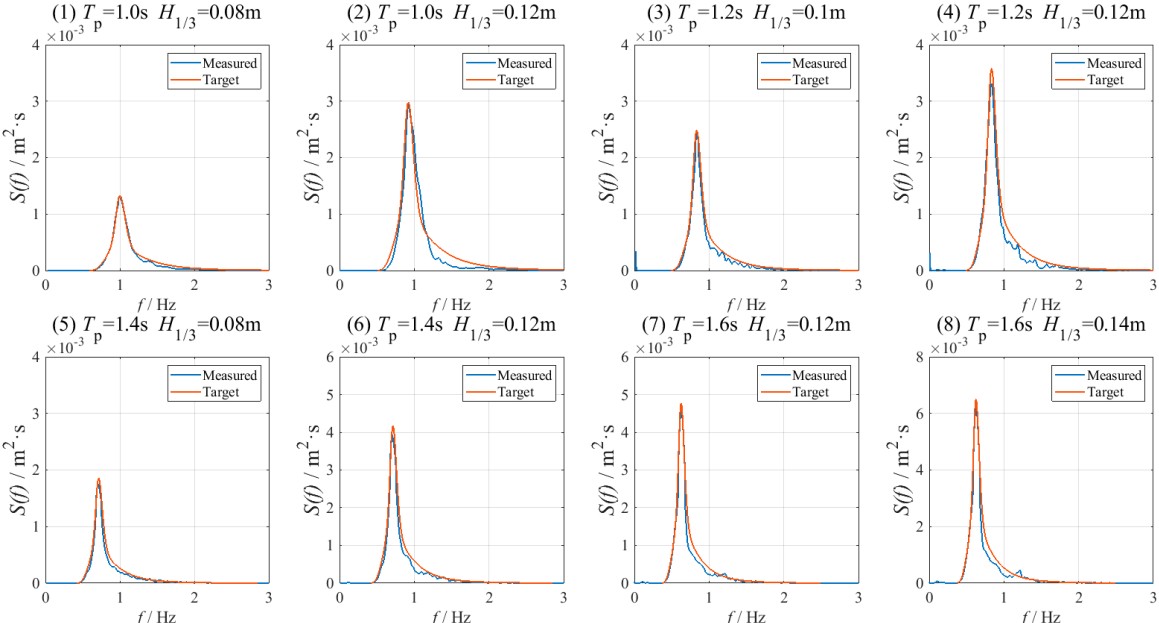

**Figure 6.** Comparisons of measured and target spectra for irregular waves with different parameters.

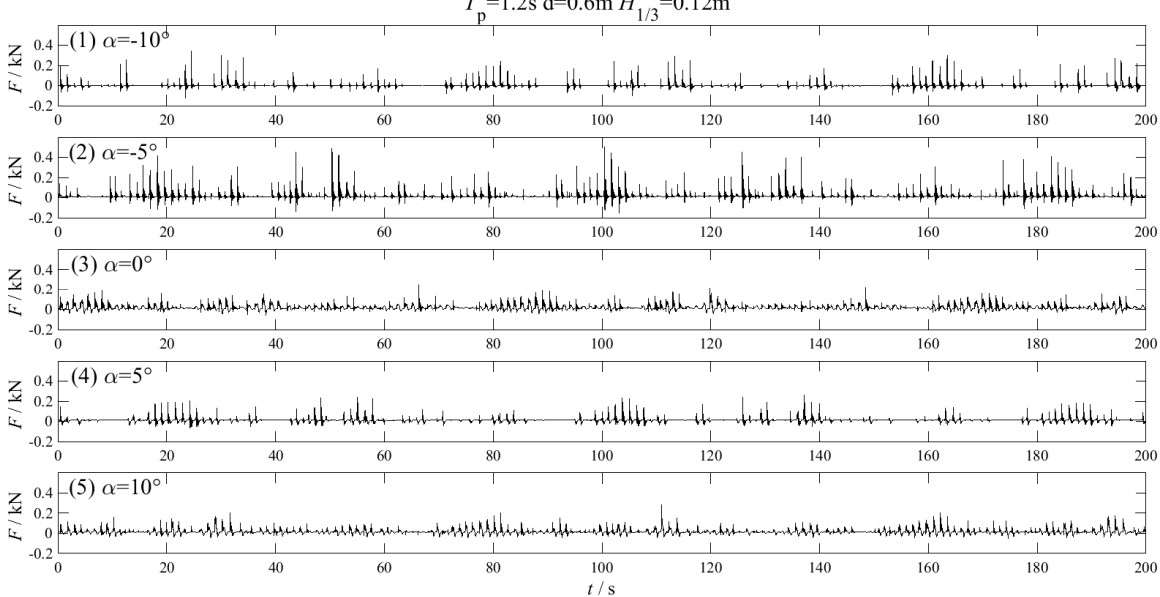

**Figure 7.** Time histories of the wave uplift forces in physical experiments with different inclined angles: $T_\mathrm{p}$ = 1.2 s, $H_{1/3}$ = 0.12 m.

A significant uplift force is defined as the average value of the largest one-third peak uplift force, which is applied commonly to represent the extreme wave impact [25]. Figure 8 presents the changes in wave uplift forces on the underside as a result of both incident wave height and relative clearance of the structure. For an inclined deck with a constant incident wave height, the uplift forces increase with decreasing relative clearance. The response of the uplift force varies as $B/Ls$ increases. For a given relative clearance in the range of $0.115 < B/Ls < 0.15$, an orderly decrease in the uplift force is seen as the relative deck width increases, and in the range of $0.15 < B/Ls < 0.325$, the values of the uplift force

increase as the relative deck width increases. The uplift forces for $\alpha = 5°$ and $\alpha = 10°$ present higher values than that for other angles, and this finding is mainly attributed to the severer inclined angles. In addition, significant deformations of the wave profile and more energy dissipation on the structure can be induced in the two cases.

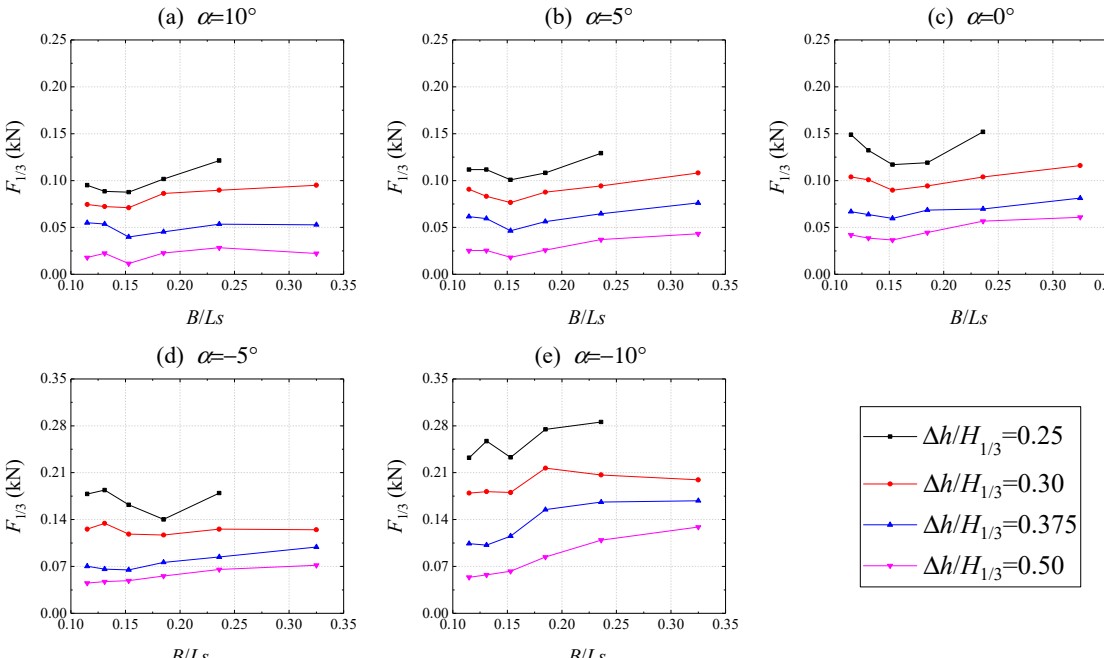

**Figure 8.** Wave uplift forces in physical experiments with respect to the relative deck width $B/Ls$ for different relative clearances. $\alpha =$ (**a**) 10°, (**b**) 5°, (**c**) 0°, (**d**) −5°, (**e**) −10°.

The uplift forces of irregular waves for the inclined angles $\alpha$ increase slowly and nonlinearly with the increase in $B/Ls$ (see Figure 9). The variation in uplift forces for a given inclined angle is rather small, but for different inclined angles, especially $\alpha = 10°$, the uplift force values are significantly larger than others. It is obvious that the influence of the inclined angle of the plate plays an important role in determining the uplift force.

### 3.3. Prediction Model of Wave Uplift Force

Previous study [26,27] and studies above show that the wave uplift force is the result of several influencing factors, including variables related to the process of wave uplift on the deck: wave dynamics, mixture of liquid–air, the relative inclined deck width $B/L_S$, relative wave height $\Delta h/H_{1/3}$, and the plate's inclined angle $\alpha$. Since the process of slamming is rather complicated, an attempt was made to develop an empirical formula for the prediction of the overall wave uplift force of the inclined deck. The empirical wave uplift force $F_{exp}$ is influenced by a number of independent variables, e.g., $H_{1/3}$, $L_s$, $\Delta h$, $d$, $B$, and $\alpha$. The relationship can be expressed in the functional form as

$$F_{exp} = f(H_{1/3}, Ls, \Delta h, d, B, \alpha) \tag{16}$$

The analysis produces four groups as follows:

$$F_{exp} = f(\frac{H_{1/3}}{Ls}, \frac{\Delta h}{d}, \frac{B}{Ls}, \frac{\pi\alpha}{180}) \tag{17}$$

Based on the Empirical formulas from predecessors [28,29], curve fitting method was applied to put forward to a new formula which involve the influence the inclined angel of structure, and the final empirical formula is expressed as

$$F_{emp} = 0.078 \cdot x_1 [0.0415 \cdot (x_2 - x_3) + 0.146] \cdot x_4 \tag{18}$$

where

$$x_1 = \rho \cdot g \cdot H_{1/3} \cdot \frac{H_{1/3}}{Ls} \tag{19}$$

$$x_2 = \ln(1.56 \cdot \left(\frac{H_{1/3}}{Ls}\right)^2 - 0.32 \cdot \frac{H_{1/3}}{Ls} + 0.03) \tag{20}$$

$$x_3 = 29.12 \cdot \frac{\Delta h}{d} \cdot \frac{B}{Ls} + 10.54 \cdot \frac{B}{Ls} - 6.43 \tag{21}$$

$$x_4 = e^{3\pi\alpha/180} \tag{22}$$

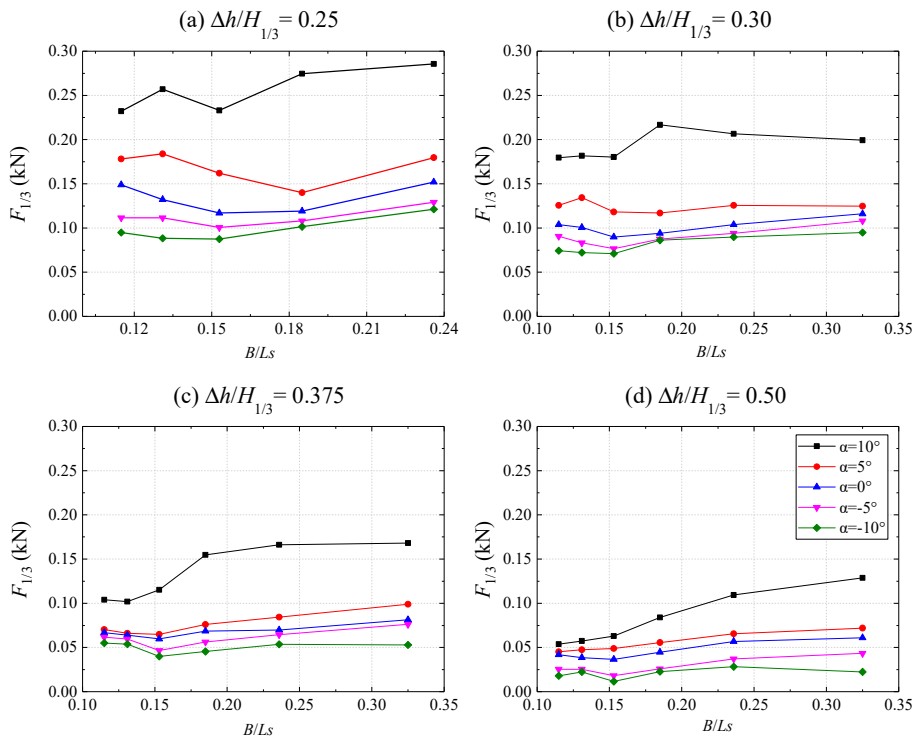

**Figure 9.** Wave uplift forces in physical experiments with respect to the relative deck width $B/Ls$ for different inclined angles. $\Delta h/H_{1/3}$ = (**a**) 0.25, (**b**) 0.30, (**c**) 0.375, (**d**) 0.50.

Figure 10 shows the validation of the predicted results obtained from the empirical function by comparison with the experimental results. The x-axis contains the predicted values calculated from Equation (18), and y-axis contains the measured values. To facilitate comparison, the line for $y = x$ is plotted in Figure 10. It can be seen that the values evenly disperse close to the $y = x$ line. To quantify the linear correlation between the two set of values, two different parameters are applied: correlation coefficient $R^2$ and standard deviation $\sigma$, and the results are shown in Table 2 as well. Relating the results with points positions is clear. For example, for $\alpha = -10°$, most points are near the auxiliary line but deviate a little above, so the correlation coefficient $R^2$ is less than other conditions. In general, to the empirical model can accurately predict the uplift force on the deck with an averaged $R^2 = 0.84$ and averaged $\sigma = 0.024$. At last, a Single factor sensitivity analysis is applied to figure out which of the parameters given in Equation (17) are governing the wave impact forces. The benchmark case is chosen

as follows: $H_{1/3}$ = 0.09 m, $Ls$ = 3.0 m, $\Delta h$ = 0.03 m, $d$ = 0.6 m, $B$ = 0.5 m, $\alpha$ = 0°. All the parameters will float up and down 5% based on the benchmark case, and each value will cause a corresponding value $\Delta F_{emp}$. The sensitive factor $\varepsilon$ is defined as the ratio of $\Delta F_{emp}$ and changeable percentages of the parameter. The results are listed in Table 3. It can be clearly observed that the wave height $H_{1/3}$ is the governing parameter, followed by inclined angle $\alpha$, which also has a great influence on the uplift force.

**Table 2.** The coefficient of determination $R^2$ and standard deviation $\sigma$ for different inclined angles.

| Angle | −10° | −5° | 0° | 5° | 10° |
|---|---|---|---|---|---|
| $R^2$ | 0.78 | 0.85 | 0.81 | 0.85 | 0.89 |
| $\sigma$ (×10$^{-2}$) | 2.45 | 1.60 | 2.40 | 2.85 | 2.48 |

**Table 3.** The sensitive factor for different parameters.

| Sensitive Factor | $H_{1/3}$ | $Ls$ | $\Delta h$ | $d$ | $B$ | $\alpha$ |
|---|---|---|---|---|---|---|
| ±5% | 0.69 | −0.16 | −0.02 | 0.02 | −0.17 | 0.37 |

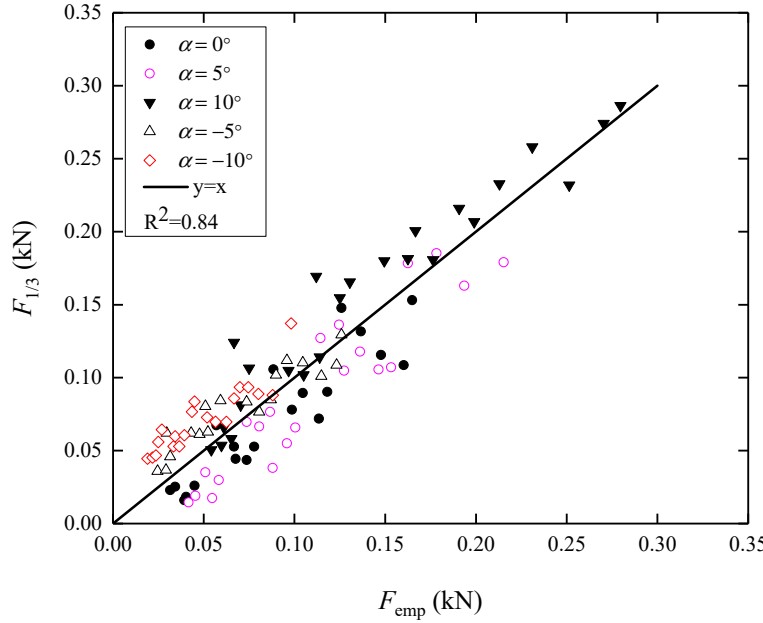

**Figure 10.** Comparison between the predicted values and experimental results of the significant uplift force.

## 4. Conclusions

Experimental analysis of slamming impact problems between waves and an inclined plate was performed and validated using a newly established NWT. A validation of wave generation in the NWT was conducted for a regular wave, where $H$ = 0.08 m. The comparison showed good agreement between experimental measurements and numerical results of the impact pressure detected at the bottom of the plate. Thus, the physical wave tank was properly set up for investigating the wave–plate interaction.

Deck models with inclined angles were applied in a physical wave tank for assessing irregular waves with different parameters to estimate the wave uplift force. Adjustable variations were as follows: inclined angle $\alpha$, relative deck width $B/L_s$, and relative clearance $\Delta h/H_{1/3}$. The results led to the following observations. First, for an inclined deck with a constant relative clearance, the wave uplift force decreases with a rise in the relative clearance $\Delta h/H_{1/3}$. Second, for a given relative clearance, the value of the uplift force increases as the relative deck width $B/L_s$ increases. Lastly, the uplift forces of irregular waves for inclined angles $\alpha$ increase slowly and nonlinearly with the increase in

$B/L_s$. For different inclined angles, especially $\alpha = 10°$, the values of uplift forces become significantly larger than those in other conditions. It was obvious that the influence of the inclined angle of the structure is pronounced, and extra attention should be paid to this aspect as a precaution for platforms when large inclinations are involved. Slamming is a phenomenon which has strong nonlinearity and randomness. For a combination of small wave period $T$ and large wave height $H$, the wave steepness will increase gradually, and waves will easily break, which is common in slamming progress. Therefore, slamming is always accompanied by physical phenomena, such as rolling, breaking, and aerated waves. This should be the dominant cause for impact values to become larger or smaller or irregular.

A predictive empirical formula of wave uplift force on an inclined deck was proposed. The model has a coefficient of determination $R^2 = 0.84$. An equation was built to establish the relationships between wave uplift force and dimensionless variables. The predicted results obtained from the empirical model were compared with the experimental results. The equation is simple and particularly useful for providing a quick estimation of the overall wave uplift force of an inclined deck.

**Author Contributions:** Z.M. conducted the experiments and wrote the original draft paper; T.Z. conceived and designed the numerical model; N.R. analyzed the experimental data; G.Z. motivated this paper with good ideas and funding support.

**Funding:** This research was supported by the Fundamental Research Funds for the Central Universities, National and Province Natural Science Foundation of China (Grant NO. 51709040, 201601056). The financial supports are greatly acknowledged.

**Conflicts of Interest:** The authors declare no conflict of interest.

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
