# Peer review of "A Comprehensive Study of the Wave Impact Loads on an Inclined Plate"

_jmse, doi:10.3390/jmse7040103_

Round 1

Reviewer 1 Report

Wave impact loads on an inclined plate are recorded in a physical and numerical model test. The parameters of the study are well chosen, so that a discussion of different dimensionless values enables a better understanding of physical process. The results from the numerical model test are validated against physical model tests. Finally, a formula based on physical model tests with irregular waves is provided to predict the wave uplift forces for inclined plates with a goodness of fit of R2=0.84.

Comments for the authors:

General comments:

1.       The authors describe physical model tests for wave spectra (JONSWAP) and compare regular waves in the numerical part of the paper. Please specify this procedure.

2.       The authors introduce a numerical model which is able to reproduce wave impact pressures on inclined plates for regular waves. However, there is no further application of the numerical model to the paper. The authors could provide the same parameter study as given in section 3.2, but then for wider ranges of the dimensionless parameters. This discussion could provide information/tendencies about marginal conditions.

3.       The quality of the figures can be optimized. Use vector graphics. The font size of Figure 6 to 9 is too small.

4.       Section 2.2: How many waves are generated for each spectrum? Or did the authors choose a duration of the tests? This information is required to analyze the impact pressures appropriate. The information could be included in table 1.

5.       Section 4: The authors sum up their observations of the study and the analysis. However, in many cases a rationale is missing. What is the reason for values to become larger or smaller? Which physical processes are in charge for the observed effects?

Detailed comments:

Line       Comment

9:            Change “Water impact…” to “Water wave impact…”

99:          Figure 1: Please include the position of the model relative to the wave maker and the flume’s side wall. Also add the plate dimensions. Add the position of the instrumentation (wave gauges) to the sketch.

129:       Are the presented parameters describe the target parameter range (e.g. ∆h/H1/3) or the measured ones? It seem like only the target values are given as the values for test No. 1 to 6 for instance are exactly the same which is quite unusual for physical experiments. Furthermore, if such detailed information about the tests are given, the target value (e.g. mean. and max. wave impact) is from interest. If you only provide the target values the range of the target parameters might be sufficient (e.g. 0.25 < ∆h/H1/3 < 0.5, -10° < α <10°, …).

197:       “The relative error of wave height is 5.5% …”. What is the reason for this error? Is the physically or the numerically derived value more reliable?

200: Where is the location (point/wave gauge) relative to the wave board or to the plate located the authors compare the analytical and numerical solution? Is the 12th point the 12th pressure sensor? If yes, how was the wave height measured in the physical model directly under the sensor?

207:       Are the authors able to compensate the difference in the wave heights between physical and numerical model by a correction function? There should be a correlation between wave height and measured impact pressure that can be derived from the data recorded in the physical model tests.

217:       Figure 5: The scatter of the peaks from wave to wave for regular waves is meaningful. What is the mean value of all recorded impacts in the physical and numerical model test? How do these values differ? What is the peak rising time in the physical model test and in the numerical model test?

226:       “… are in good agreement …”: What is a good agreement? Please quantify the optical good agreement with the standard deviation and the goodness of fit. Is the good agreement the same for all frequencies of the presented spectrum?

228:       “… the wave uplift force of the deck was calculated by integration …”: please provide the corresponding equation that derives the force from the measured pressures.

237:       Why did the authors define the average value of the largest one-third peak uplift force as “significant” uplift? Is this the relevant design case? How about the influence of the maximum peak in relation? Please provide a reference to the literature. In general, a comparison to findings from literature to very similar cases is missing in the analysis section.

273:       Something went wrong with the formula.

281:       Figure 10: The goodness of fit can be reduced from 0.8351 to 0.84. Please add the standard deviation. For an easier quantification of the errors the authors could add e.g. the 90% percentile lines to the graph. The authors should add a deeper discussion for Figure 10. E.g. Uplift forces for α=10° are under predicted by the formula, uplift forces for α=10° are over predicted. The authors could add an additional table that provides the standard deviation for each plat inclination.

Author Response

The authors would like to thank the reviewers for their constructive comments which have helped to improve the manuscript significantly. Please open the revision mode in the revised manuscript to correspond to the line number. Our responses to the questions are provided in word (typed here in red) following each question.

Reviewer 2 Report

Line 157, page 5/12

Eq (1) ® Eq (6)

Eq. 12 and 13, along with Line 164, page 5/12

Please avoid using the same symbol for two different parameters (C for surface tension coefficient and for phase velocity of the target wave).

Eq. 13, page 5/12

What is the physical meaning of „A“?

Fig. 5, page 8/12

What is the numerical model time step according to CFL=0.5, and is it possible to relate the difference between measured and modelled results (presented in figure 5 ) with differences in pressure transducer sampling rate (1000 Hz) and calculation time step  ?

Fig. 6, page 8/12

There is no need to present comparison of physical and numerical wave spectra. After Fig. 6 there is no further results and discussion regarding numerical modelling.

Fig. 7, 8, 9, pages 9/12 and 10/12

Within the figure capture one should indicate that presented results are obtained only via physical model testing.

Please, increase used font.

Please, provide the values of maximum measured pressure in Figs. 8 and 9 (secondary „Y“ axes can be used). A few information about the position of maximum pressure occurrence would be of great help for the comprehension of analysed process.

page 10/12

Please compare the results obtained with eq. 16 for the case of horizontal plate with the results that one would calculate using the very same wave and plate geometry data set and engineering empirical equations according to:

-          Cuomo, G., Tirindelli, M., Allsop, N.W.H.: Wave-in-deck loads on exposed jetties, Coastal Engineering, 54 (2007) 9, pp. 657-679.

-          USACE: Coastal Engineering Manual _ Part 6, Chapter 5, Fundamentals of Design. 2006.

-          Det Norske Veritas: Recommended Practice – Environmental Conditions and Environmental Loads, DNV-RP-C205, 2010.

-          Kaplan, P., Murray J.J., Yu W.C.: Theoretical Analysis of Wave Impact Forces on Platform Deck Structures, Volume 1-A Offshore Technology, OMAE - Offshore Mechanics and Arctic Engineering Conference, Copenhagen, pp. 189-198, 1995.

Author Response

Dear editor and reviewers, The authors would like to thank the reviewers for their constructive comments which have helped to improve the manuscript significantly. Please open the revision mode in the revised manuscript to correspond to the line number. Our responses to the questions are provided in word(typed here in red) following each question.

Round 2

Reviewer 1 Report

The reviewer thanks the authors for clear replies to the given questions and for incorporating some suggestions. Based on the improvements by the authors additional questions arise which are given in the following:

General comments:

1.                   In the discussion, a reference and comparison to the diverse findings from literature is missing:

a.       e.g. a comparison with existing studies ([5], [15], …)

b.      A statement regarding the influence of “the compressibility of the water, air cushions, air bubbles, and even hydro-elasticity” on the presented results as stated by [11].

c.       Did the authors identify a “substantial similarity between the mechanisms of wave impact on horizontal platforms and those on vertical barriers” as [14],[15]?

2.                   Wave impact loads are addressed by the title of the article. In the paper the authors sometimes address forces, sometimes address pressures. It is correctly stated that pressure (N/m²) is force (N) per area (m²) but the terms are mixed in the paper. E.g. line 284 stats uplift forces in the following table, but impact pressures are given. Please correct this for the whole document and follow a consistent procedure.

3.                   In the references, double numbering is used from reference 3 on.

Detailed comments:

Line       Comment

223         A spacing is missing in the figure caption “ofthree” à “of three”.

243         Change “an dangerous area” to “a dangerous area”.

246         The authors refer a “good agreement with experimental results at most measure points”. Please specify where you have a mismatch and account on the reasons.

247         In Table 1 the authors provide uplift forces or impact pressures? Furthermore, please specify in the caption of the table if these are maximum or mean values.

273         The font size of Figure 6 is too small. Please increase it in analogy to Figure 8.

275         In Figure 5 the axis title are confusing. Did the authors mean p [kPa] and t [s] instead of p /kPa and t/s? Please proof for all figures.

391         Figure 10: Regarding the author’s comment in the 1st review reply standard deviations should be given in a following table, which is not included in the provided paper for review.

403         The authors provide the three major findings and of the paper. It would be helpful to bring them in relation to findings for non-inclined decks from the presented study as well as from literature. This would highlight the meaning of the present study for design cases.

417         Simplify R²=0.8351 to 0.84.

417          “An equation was built to establish the relationships between wave uplift force and dimensionless variables.” Please address, which of the dimensionless parameters given in Eq. (17) are governing the wave impact forces and which only have a minor effect.

Author Response

The authors would like to thank the reviewers for their constructive comments in the second round. The revised version of the article has been completed after taking 6 days nearly, please open the revision mode in the 2th revised manuscript to correspond to the line number. Our responses to the questions are provided below (typed here in red) following each question:

Reviewer 2 Report

I believe the manuscript has been significantly improved and now warrants publication in JMSE.

Author Response

Thanks to the reviser’s suggestion.